# PRESTO: A Framework for Orchestrating System States and Test Cases for Bash Script Verification

## Abstract

Bash is a widely used scripting language for automating system and cloud tasks, but its reliance on implicit preconditions—such as environment variables, file paths, and tool availability—makes it error-prone, especially when scripts are generated by large language models (LLMs). While LLMs have demonstrated promising capabilities in translating natural language to Bash scripts, the lack of reliable evaluation methods and test coverage hampers their practical utility. We introduce PRESTO, a modular framework for **Prerequisite-aware Script Testing and Orchestration**, designed to assess and refine Bash scripts through execution-driven feedback loops. PRESTO automatically infers required preconditions, synthesizes minimal reproducible environments, generates targeted test cases, and evaluates the behavior of both LLM-generated and human-authored Bash scripts in a sandboxed execution environment. Upon failure, an iterative refinement cycle—driven by LLMs—updates the script, environment setup, or test harness until correctness is restored. Our experiments on two public benchmarks show that PRESTO significantly improves correctness, debugging efficiency and reliability compared to static or heuristic methods. Unlike reference-based metrics, PRESTO operates without requiring gold-standard references, making it suitable for real-world deployment scenarios. This positions PRESTO as a practical solution for production-ready script generation.

## 1 Introduction

Bash scripting plays a critical role in modern computing infrastructure. From system administration to cloud DevOps and Site Reliability Engineering (SRE), Bash scripts are routinely used to automate tasks such as service restarts, log analysis, resource monitoring, and deployment pipelines. In production systems, these scripts are often the first line of defense in mitigating failures Mayank Agarwal & White (2021); Xi Victoria Lin & Ernst (2018); an incorrect or brittle script can increase mean time to resolve (MTTR) and introduce costly downtime Atlassian (2018); Balbix (2025); Puli (2025). Despite their importance, Bash scripts remain challenging to develop and verify. Their execution depends heavily on implicit assumptions such as file system state, environment variables, installed utilities, and access permissions. These assumptions —referred to as **prerequisites**— are rarely documented, and lead to runtime failures or silent logic errors when violated Tips (2021).

Large language models (LLMs) have recently been applied to Bash code generation from natural language prompts, enabling non-experts to automate tasks and assisting experts in writing scripts faster Aggarwal et al. (2024); Yang et al. (2023); Westenfelder et al. (2024). However, evaluating and refining these scripts is inherently more complex than assessing programs in languages like Python or Java Chatterjee et al. (2025). Simple syntactic checks or string-based heuristics fail to capture semantic equivalence (e.g., multiple disk-usage commands that yield equivalent results with different formats). The problem is compounded when reference-based evaluation metrics (e.g. BLEU, Crystal-BLEU) are used: these metrics measure token overlap but not runtime semantics, leading to inflated scores despite program invalidity Liu et al. (2023); Yang et al. (2023). As a result, generated Bash scripts often appear plausible but fail during execution, leading to undetected bugs in automated workflows. Even execution-based baselines are limited: without explicit modeling of prerequisites and robust test cases, failures are often misattributed, and incorrect scripts may

slip through undetected. This lack of reliable evaluation makes it difficult to trust automatically generated Bash scripts in high-stakes environments such as SRE workflows.

In this work, we present PRESTO, a prerequisite-aware evaluation framework for Bash scripts. Given a natural language task, PRESTO infers prerequisite steps required to set up the execution environment, generates corresponding test cases that capture task-level semantics, and executes the main bash script in a sandbox. Failures are analyzed by a feedback-driven refinement loop that iteratively corrects the prerequisite or test scripts until the environment and evaluation are stable. Importantly, prerequisite and test case generation are performed without access to the main script, ensuring that they remain faithful to the input task rather than overfitting to a particular implementation. Only at the execution stage is the main script introduced. This design enables PRESTO to deliver robust, reliable evaluation of Bash scripts.

Although PRESTO is an evaluation framework at its core, its impact extends further. By providing accurate and semantically grounded feedback signals, PRESTO enables downstream refinement of Bash scripts themselves. Thus, the same mechanism that strengthens evaluation also improves script synthesis: LLM-generated scripts can be iteratively repaired using PRESTO's judgments. We validate PRESTO on NL2Bash-EABench Aggarwal et al. (2024) and InterCode-Corrections Yang et al. (2023) benchmarks, demonstrating two key results: (i) prerequisite-aware evaluation significantly improves the reliability of Bash script validation, and (ii) PRESTO's feedback enables measurable improvements in end-to-end Bash code generation accuracy. Together, these contributions position PRESTO as both a principled evaluation framework and a catalyst for more trustworthy script generation in real-world automation workflows.

## 2 RELATED WORK

The task of code generation from natural language has seen significant progress with large language models (LLMs) such as GPT-4, LLaMA, Deepseek, and Mistral (Touvron et al., 2023a;b; Guo et al., 2024). Beyond model architecture, recent efforts enhance generation quality through post-processing. CodeT (Chen et al., 2023a) pairs code with test cases and uses dual-agreement filtering, while coder-reviewer (Zhang et al., 2022) and CodeGen (Nijkamp & Others, 2022) apply ranking heuristics. A particularly promising class of techniques is self-debugging, where LLMs refine outputs using execution feedback. Methods like Self-Debug (Xinyun Chen & Zhou, 2023), LDB (Wang & Shang1, 2024), AutoDebug (Jiang et al., 2024), and broader evaluations by Adnan et al. (Muntasir Adnan & Kuhn, 2023) demonstrate the scalability of this approach without increasing sampling cost (Xinyun Chen & Zhou, 2023; Yang et al., 2024; Dong et al., 2023; Huang et al., 2023b).

In the NL2SH domain, early evaluations used string similarity and functional equivalence heuristics (Mayank Agarwal & White, 2021; Xi Victoria Lin & Ernst, 2018), which lacked semantic depth. InterCode-Bash (Yang et al., 2023) introduced execution-based validation using Docker isolation and side-effect comparisons, though it may misjudge semantically equivalent commands with divergent outputs. Meanwhile, non-execution metrics like CodeBLEU (Shuo Ren & Ma, 2020), CodeBERT (Zhangyin Feng & Zhou, 2020), and CrystalBLEU (Eghbali & Pradel, 2022) compare predictions to references without code execution. Hybrid techniques such as CodeSift (Aggarwal et al., 2024) use LLMs for textual and semantic comparisons but rely on accurate NL translations and reference access.

InterCode-ALFA (Westenfelder et al., 2024) advances hybrid evaluation by combining functional correctness with LLM-based semantic scoring. However, it still depends on predefined references, limiting its flexibility in reference-less scenarios like Bash command generation. In contrast, our approach eliminates this dependency through automated test generation and self-refinement.

Recent work like InverseCoder (Yutong Wu & Chen, 2024) proposes a self-bootstrapping method to instruction-tune LLMs without relying on closed-source models. By generating instructions from existing code and retraining iteratively, models such as CodeLlama-Python and DeepSeek-Coder achieve gains on benchmarks like HumanEval(+), MBPP(+), and DS-1000. Building on this, our framework unifies instruction tuning, test generation, execution validation, and iterative refinement to enable reference-less, robust self-debugging in both code generation and NL2SH tasks. Recent multi-agent frameworks Chen et al. (2023b); Hong et al. (2024); Huang et al. (2023a) target code

generation but rely on given tests, limiting their applicability to Bash evaluation; AgentCoder Huang et al. (2023a), the only one with an evaluator agent, serves as the most relevant baseline for comparison with PRESTO .

## 3 METHODOLOGY

We present the overall system architecture followed by a detailed description of PRESTO, the core contribution of this work. The system is designed to support the full lifecycle of Bash script handling: script generation, evaluation, and refinement. Among these, the novelty lies in PRESTO, a prerequisite-aware evaluation framework that systematically validates scripts under realistic execution environments.

### 3.1 SYSTEM OVERVIEW

The overall system operates as an end-to-end pipeline orchestrated by specialized agents:

- **Code Generation:** Generates Bash script $M$ from the natural language task description $S$ using large language models. Detailed prompts are listed in Appendix B
- **Code Evaluation via PRESTO:** Assesses $M$'s correctness in a prerequisite-aware manner (detailed below). If PRESTO deems $M$ correct, the pipeline succeeds.
- **Code Refinement** If PRESTO outputs that $M$ as incorrect, the Script Refiner agent activates to refine $M$ using PRESTO's feedback (e.g., error traces).

This architecture enables reliable code generation by integrating generation, evaluation, and targeted refinement.

We now dive deep into the core novelty of this paper: PRESTO , which performs prerequisite-aware script evaluation that powers the above system. Unlike prior methods that rely on syntactic similarity metrics, reference based comparisons or heuristic grading, PRESTO explicitly models the implicit environmental dependencies present in system related tasks and validates scripts through execution based feedback loops. This enables robust validation even in the absence of ground truth reference scripts, while also providing a reliable and trustworthy explanation for why a certain script is correct or incorrect.

### 3.2 PRESTO: PRErequisite-aware Script Testing and Orchestration

PRESTO'S design is motivated by the observation that system-level scripts generally need specific environment configurations to execute successfully, for example - file system states, process availability, permission configurations etc. PRESTO addresses the critical gap in existing state-of-the-art code evaluation framworks by explicitly modeling and validating these implicit prerequisites that are required for correct script execution for system related tasks. Unlike SOTA frameworks such as Huang et al. (2023a) which perform testcase generation in a one-shot approach for evaluation, the core novelty of PRESTO lies in the fact that it decomposes the evaluation into three components - (i) environmental prerequisite setup, (ii) targeted testcase generation for functionality validation and (iii) iterative refinement through execution driven feedback.

Figure 1 illustrates the overall pipeline of PRESTO with the help of an example: starting from a natural language task $S$, the system generates prerequisite steps ($P_s$) and test case steps ($T_s$), converts them into executable scripts, and finally uses these scripts to evaluate the main script $M$ inside a sandboxed environment. The figure highlights the sequential flow from task $\rightarrow$ step planning $\rightarrow$ code generation $\rightarrow$ execution $\rightarrow$ refinement, showing concretely how prerequisites and test cases are aligned to the input task.

PRESTO begins by jointly generating natural language steps for both prerequisites ($P_s$) and test cases ($T_s$) from the prompt $S$. This joint generation ensures alignment between the environment setup and validation criteria, preventing mismatches that could lead to false positives or negatives.

Once the steps are generated, dedicated code generation agents translate them into executable scripts: the prerequisite script $P$ from $P_s$, and the test script $T$ from $T_s$ (with awareness of $P$ for consistency).

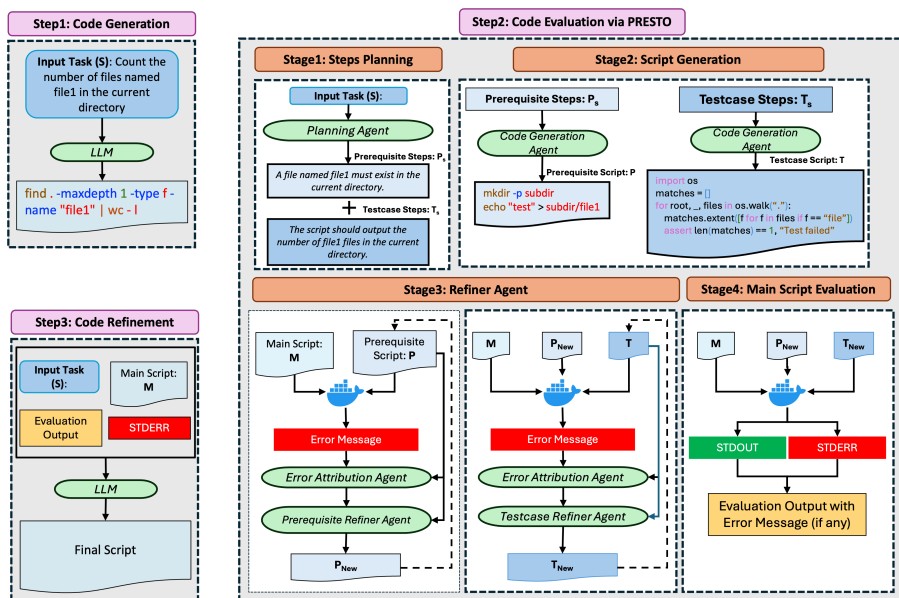

Figure 1: Overview of the pipeline. The figure spans setup, generation, and evaluation phases, showing the interplay between natural language prompt, code generation models, refinement loops, and execution-based validation.

Next, PRESTO executes the prerequisite script $P$ followed by the main script $M$ inside the sandboxed environment. If execution fails (evidenced by a non-zero exit code or stderr), the **Error Attribution Agent**—an LLM acting as a reviewer—examines the error traces and exit codes (prompt template: `Given error: [error] and exit code: [e], classify as: prerequisite issue / test issue / main script bug`). If the error is classified as a *prerequisite issue* (for instance, missing files, directory mismatches, or unset environment variables that prevent $M$ from running), the **Prerequisite Refiner Agent** is invoked to regenerate the Prerequisite script $P$.

**Important design note:** during this refinement step, only the current prerequisite script $P$, the task description $S$, and the observed error diagnostics are provided to the refiner. The main script $M$ is *never passed as input* to this agent. This deliberate restriction ensures that prerequisite corrections remain aligned to the *intended task specification* rather than being biased toward the particular implementation of $M$. In this way, prerequisites reflect what is required for any correct solution of $S$, rather than being overfit to the idiosyncrasies of a potentially buggy main script.

Once the prerequisite is stabilized (i.e., both $P$ and $M$ execute successfully without errors ), PRESTO proceeds to evaluate the full triplet $\{P, M, T\}$ in the sandbox. If the test case $T$ fails, the Error Attribution Agent classifies the error. When the failure is attributed to flawed test logic (e.g., incorrect assertions, mismatched expected outputs), the **Test Case Refiner Agent** is invoked to iteratively correct $T$.

**Important design note:** during test case refinement, the refiner only receives the current prerequisite script $P$, the test script $T$, the natural language task $S$, and the associated error diagnostics. The main script $M$ is deliberately withheld from this process. If $M$ were included, there is a high risk that the refiner could generate trivial or biased tests that always declare $M$ as correct, leading to false positives. By isolating test generation from $M$, we ensure that test cases are aligned with the original task specification and validation logic rather than the quirks of a particular implementation. In fact, none of the planning or code generation agents in PRESTO have access to $M$; the main script is introduced only at the execution stage. This strict separation prevents evaluation leakage and enforces the robustness of the testing process.

After both the prerequisite script ($P$) and test case script ($T$) have been refined to stability (i.e., no further errors are detected during execution), PRESTO proceeds to evaluate the main script ($M$) against the validated environment and test logic. At this stage, the output of PRESTO is twofold: (i)

a final verdict on whether $M$ satisfies the task description, and (ii) a structured explanation derived directly from the failed test cases whenever the verdict is negative. This explanation pinpoints which test conditions failed, along with corresponding error messages, thereby providing a transparent diagnostic trace. The structured explanation is then passed to the downstream refinement loop for the main script.

### 3.3 MAIN SCRIPT REFINEMENT

PRESTO 's decision directly informs downstream refinement. If $M$ is deemed correct, the pipeline terminates successfully. If $M$ is judged incorrect (e.g., persistent failures attributed to $M$), the feedback signals from PRESTO (error traces, failed test message, diagnostics) are passed to an LLM-based Script Refiner agent. This agent refines $M$ with access to $S + M$ and feedback signals. This process continues iteratively until a correct script is produced or the maximum refinement budget is reached. Note that, during this step neither the Prerequisite code $P$ nor test case code $T$ are refined.

## 4 EXPERIMENTAL SETUP

### 4.1 DATASETS

We evaluate PRESTO on two complementary benchmarks that capture distinct challenges in natural language–to–Bash (NL2Bash) translation and evaluation.

NL2Bash-EABench Aggarwal et al. (2024) is an execution-based benchmark designed to assess Bash script generation for system-related tasks. It consists of three progressively challenging suites, of which we use `bash_1` and `bash_2` (100 tasks total). These range from single-line utilities to multi-step operational scripts representative of real-world system administration. Crucially, EABench evaluates scripts by executing them in isolated containers, ensuring both syntactic validity and functional correctness—properties essential for downstream applications such as incident response and site reliability engineering.

InterCode-Corrections Westenfelder et al. (2024) consists of 193 carefully revised samples from the InterCode-Bash dataset, correcting annotation errors and forming part of the broader NL2SH-ALFA benchmark. Unlike EABench, which emphasizes execution correctness in controlled environments, InterCode-Corrections compares generated and gold scripts using multiple similarity measures (e.g., TF-IDF, embedding-based, and LLM-as-a-Judge). Following recent evidence that LLM-as-a-Judge is the most reliable proxy for functional equivalence, we adopt it for all experiments on this benchmark, particularly when closed-source models such as GPT4o are used as the judging LLMs.

### 4.2 BASELINES

For fair evaluation, we use the official execution harnesses of NL2Bash-EABench and InterCode-Corrections to obtain ground-truth labels for script correctness. These ground-truth verdicts are then used to measure and compare the accuracy of the following baselines and of PRESTO . We benchmark PRESTO against both execution-less and execution-based evaluation frameworks to provide a comprehensive comparison.

For the execution-less category, we consider two approaches: (i) **ICE-Score** Zhuo (2023): Prompts an LLM to assign correctness scores from 1–4 without execution. Following prior work, we map score 4 to "correct" and all others to "incorrect" for binary comparability, and (ii) **Direct Grading**, a simpler baseline we introduce, which prompts the LLM to directly provide a binary judgment—`correct` or `incorrect`—given the task description and generated script.

For the execution-based category, we adapt the **AgentCoder** framework (Huang et al., 2023a), originally designed for multi-agent collaboration in code generation, validation, and refinement. Since *AgentCoder* was initially developed for Python programming tasks, we adapt the test designer agent to support Bash-specific scenarios, enabling a fair comparison with PRESTO . Importantly, Agent-Coder is the best prior method for code generation that performs reference-free execution-based evaluation through testcase generation, making it the closest baseline to PRESTO .

### 4.3 MODEL AND CONFIGURATION SETTINGS

We evaluate all methods using both closed-source and open-source language models (GPT-4o (OpenAI, 2024), Llama-4 Maverick (Meta AI, 2025), Mistral Medium-3 (Mistral AI, 2025)) to demonstrate the broad applicability of our approach. For consistency and to avoid bias introduced by model switching, the same underlying LLM is used for both code generation and code evaluation in each experimental setting. This ensures that performance differences are attributable to the evaluation framework rather than to discrepancies in model capability (Zheng et al., 2023; Chatterjee et al., 2025). For all models, we set temperature = 0 to ensure deterministic outputs and reproducibility, while retaining other parameters at default settings. For PRESTO , we utilize a two-language scripting setup: (i) Bash for implementing prerequisite tasks. (ii) Python for constructing and executing test-case scripts. We additionally explore alternative combinations of scripting languages in ablation studies to assess their impact. The maximum number of refinement loops is set to 5. Analysis of refinement iterations is present in appendix A

## 5 RESULTS

We now turn to an empirical analysis of PRESTO, evaluating its performance on two benchmarks and comparing it against established baselines. Our goal is to understand how well PRESTO can evaluate Bash scripts, where its strengths and limitations lie, and how different design choices affect its effectiveness. To this end, we structure the results around a series of research questions, each addressing a specific aspect of the framework: from the accuracy of script evaluation and prerequisite identification, to the benefits of multi-language settings, the impact of feedback-driven refinement, and the downstream effect on Bash code regeneration.

| Model | Method | NL2Bash-EABench | | Intercode-Corrections | |
|---|---|---|---|---|---|
| | | Accuracy | F1-Score | Accuracy | F1-Score |
| GPT4o | *Direct Grading* | **81%** | **58.41%** | 54.4% | 45.77% |
| | *ICE-Score* | 77% | 55.6% | 44.56% | 44.34% |
| | *Agent Coder* | 31% | 30.16% | 49.74% | 39.93% |
| | *PRESTO* | 79% | 57.53% | **60.1%** | **58.9%** |
| LLama4 Maverick | *Direct Grading* | **81%** | **62.76%** | 59.58% | 50.42% |
| | *ICE-Score* | 78% | 48.28% | 46.11% | 46.11% |
| | *Agent Coder* | 39% | 38.85% | 46.11% | 37.12% |
| | *PRESTO* | **81%** | 61.57% | **63.73%** | **55%** |
| Mistral Medium | *Direct Grading* | **81%** | 57.21% | 53.84% | 41.07% |
| | *ICE-Score* | 79% | 47.61% | 45.6% | 45.54% |
| | *Agent Coder* | 41% | 40.71% | 45.6% | 36.22% |
| | *PRESTO* | **81%** | **65%** | **63.21%** | **62.42%** |

Table 1: Performance of evaluation approaches in grading generated scripts. The metrics reflects how reliably each method labels correct scripts as correct and incorrect scripts as incorrect

### 5.1 COMPARING THE ACCURACY AND RELIABILITY OF EVALUATION APPROACHES FOR BASH SCRIPTS(RQ1)

We evaluate the performance of PRESTO and other evaluation approaches in grading Bash scripts for system-related tasks. Results are reported using both macro F1 and accuracy on the two benchmarks. Since the majority of labels in both datasets are positive, macro F1 provides a more balanced measure of evaluation quality than accuracy alone. Importantly, these metrics reflect the reliability of the evaluators themselves, rather than the code generation accuracy of the underlying models.

As can be seen from table 1 Direct Grading achieves the best execution-less performance across the models, with its performance being closer to PRESTO in NL2Bash-EABench. However, the limitations of execution-less evaluation is clearly seen from the performance on Intercode-Corrections dataset which contains relatively harder tasks comparted to NL2Bash-EABench. ICE-Score, while similiar and more nuanced compared to Direct Grading, falls short in both accuracy and f1 especially in Intercode-Corrections, which indicates that numeric grading can be less reliable for non-algorithmic system related tasks.

Across all the three LLMs, AgentCoder, despite being one of the strongest execution based evaluation framework for algorithmic coding benchmarks such as MBPP and HumanEval, performs poorly on system-level Bash tasks, highlighting a generalization gap when transitioning from algorithmic

programming domain to system-level scripting environments. The performance degradation can be attributed to two major reasons - i) Lack of prerequisite awareness - unlike PRESTO 's structured approach, AgentCoder directly generates testcases without explicitly modelling or setting up the environment for the given task. ii) Absence of Iterative Refinement - AgentCoder does not have any refinement loop for testcase generation (as generating single line assert based testcases for algorithmic problems are much simplier in nature).

PRESTO consistently matches or outperforms all the baselines across models and benchmarks, highlighting its robustness and generalizability. Its prerequisite aware architecture and iterative refinement capabilities enable it to handle the complex environmental dependencies inherent in system related tasks effectively. Among the models tested, Mistral Medium 3 has the most consistent gains, outperforming GPT-4o and Llama4 Maverick across nearly all settings. Consequently, we utilize Mistral Medium 3 for all subsequent experimental analyses.

## 5.2 PREREQUISITE GENERATION ACCURACY (RQ2)

To evaluate the accuracy of the Prerequisite script generated by PRESTO we utilize the ground truth (GT) scripts. We first execute the PRESTO generated prerequisite script (P) in a blank linux docker container, which is followed by the execution of the corresponding GT script. If the prerequisite script correctly creates the prerequisite for the given task, then the GT script should execute without any errors. We mark the prerequisite script as correct if and only if both the prerequisite and GT scripts execute successfully (exit code = 0).

PRESTO demonstrates high prerequisite generation performance using Mistral Medium and LLama4 Maverick, with both of them achieving accuracy rates above 90%. The high prerequisite generation performance is crucial to the execution based evaluation approach since environmental setup failure would compromise the reliability of testcase execution results for determining the script correctness. Surprisingly, GPT-4o exhibits relatively lower prerequisite generation accuracy (around 70%), which correlates to its lower overall evaluation performance (Table 1). On detailed manual analysis, we identified that GPT-4o frequently enters extended feedback refinement loops during prerequisite generation, suggesting difficulties in (i) identifying and understanding system-specific dependencies required for the given task, (ii) analyzing the errors due to incorrect prerequisite generation, and rectifying them during the refinement loop. The performance disparity across models highlights the importance of robust prerequisite generation for reliable script evaluation and also suggests that model-specific refinements (eg. prompt engineering) may be required to achieve consistent performance across the models.

| Language | Acc | F1 | Acc | F1 |
|---|---|---|---|---|
| | NL2Bash-EABench | | Intercode-Corrections | |
| Bash_Bash | 81% | 52.28% | 56.48% | 44.72% |
| Python_Python | 82% | 61.58% | 56.48% | 44.03% |
| Bash_Python | 81% | 65.00% | 63.21% | 62.42% |

Table 2: Accuracy and F1-score of PRESTO with different combinations of scripting languages.

| Method | NL2Bash-EABench | Intercode-Corrections |
|---|---|---|
| No Refinement | 78% | 52.6% |
| w/ Direct Grading | 79% (+1) | 55.4% (+2.8) |
| w/ AgentCoder | 74% (-4) | 39.48% (-13.2) |
| w/ ICE-Score | 78% (+0) | 45.6% (-7) |
| w/ PRESTO | **83% (+5)** | **56% (+3.4)** |

Table 3: Code generation accuracy with feedback-based refinement.

## 5.3 MULTI-LANGUAGE EVALUATION (RQ3)

We investigate the impact of scripting language on PRESTO 's performance. In particular we focus on bash and python as the scripting languages since we are dealing with system related task and python is the language on which most models are proficient. We evaluate 3 distinct language combinations: (i) *Bash_Bash* - Bash for both prerequisite and testcase generation, (ii) *Python_Python* - Python for both components and (iii) *Bash_Python* - Bash for prerequisites and python for testcases.

As can be seen from table 2, *Bash_Python* exibits superior performance on both the benchmarks. The performance difference can be attributed to the complimentary nature of the two script generations. prerequisite scripts typically involve system level tasks such as file system manipulation, directory creation, permission setting etc. Bash's native integration with Unix system calls and its concise syntax for common system level operations make it optimal for environmental setup scripts.

Conversely testcase scripts often involve complex logical reasoning, data structure manipulation, assertion handling etc. to verify the given code which suits python due to its expressiveness and robust standard libraries.

## 5.4 Feedback Based Main Script Regeneration (RQ4)

For refinement, scripts flagged as incorrect by the evaluation approaches are provided to the model once with the same prompt to "correct incorrect code," along with the input task and any associated feedback. Results in (Table 3) indicate that execution-less methods demonstrate limited improvements: Direct Grading yields only +1% and +2.8% on NL2Bash-EABench and Intercode-Corrections, respectively, whereas ICE-Score shows no gain on NL2Bash and a –7% drop on Intercode. This indicates that simple binary feedback lacks sufficient granularity. .AgentCoder-based refinement suffers even larger declines (–4% and –13.2%) due to poor evaluation performance: the test case generated without any knowledge of prerequisites generally label the main script as failed so the flawed fail messages propagate into refinement, causing correct scripts to be wrongly modified. A key reason for the degradation is instruction sensitivity during refinement. The model is instructed to correct "incorrect" code. However, as observed in prior studies Huang et al. (2025); Heo et al. (2024), LLMs are highly sensitive to how instructions are framed: if asked to identify or fix errors, models may hallucinate faults in otherwise correct code, producing incorrect refinements. Conversely, when asked to justify correctness, LLMs tend to provide reasons supporting the code as written, reinforcing it as correct rather than questioning it. Consequently, ICE-Score–guided and AgentCoder-guided refinement can incorrectly modify correct code, leading to a net drop in final accuracy.

In contrast, feedback from PRESTO leads to the highest improvements, raising accuracy to 83% on NL2Bash (+5%) and 56% on Intercode (+3.4%). By flagging incorrect cases with high precision and providing meaningful, execution-based insights, PRESTO delivers actionable feedback that effectively guides the correction of erroneous scripts. This demonstrates that robust and accurate evaluation is critical for successful refinement in system-level code generation tasks.

## 5.5 Analyzing role of Prerequisite and Test Case Refinement in Presto (RQ5)

To assess the contribution of execution-driven iterative refinement, we illustrate representative evaluation traces in Figure 2. These cases highlight how PRESTO corrects errors in environment prerequisites and test cases, as well as where it still fails due to coverage gaps or semantic mismatches.

**Example 1.** Both the prerequisite and test case are correct in the first attempt. PRESTO executes the main script without intervention, and its judgment matches the oracle. This represents the ideal case where high-quality generation requires no refinement.

**Example 2.** The initial prerequisite is incomplete—it creates only one directory instead of both `dir1` and `dir2` and omits the required file. Consequently, the script fails. The prerequisite Refiner Agent identifies the setup error and, in the second iteration, produces a corrected environment script. With the revised setup, PRESTO verifies the correctness of the main script, aligning with the oracle.

**Example 3.** The first generated test case encodes an incorrect MD5 hash due to a newline misinterpretation. PRESTO initially flags the script as wrong. The Test Case Refiner Agent corrects the expected hash, enabling PRESTO to reclassify the script as correct. This illustrates how targeted refinement can resolve subtle mismatches in verification logic.

**Example 4 (Coverage Gap).** In this case, PRESTO incorrectly labels the script as correct, while the ground-truth evaluation marks it as incorrect. The discrepancy arises because the generated prerequisite does not include a scenario mixing files and subdirectories. As a result, the edge case—where the script wrongly counts the current directory—is never tested. This reflects a *coverage gap* in environment generation: when generated setups fail to capture critical edge conditions, PRESTO may deliver false positives.

**Example 5 (Semantic Mismatch).** Another failure occurs when PRESTO 's test only checks for the existence of a group named `hackers`, without verifying that it is specifically a *system group*. Since this refinement error recurs across all models, it highlights a *semantic mismatch* between natural language task specifications and the LLM's generated test logic. Here, insufficient grounding in

| Task | Main Script | Steps Generation | Prerequisite Generation | Testcase Generation | Prerequisite Refinement | Testcase refinement | Evaluation |
|---|---|---|---|---|---|---|---|
| Rename file a.txt to b.json in the current directory | `#!/bin/bash`
`mv a.txt b.json` | # prerequisite: Ensure that a file named a.txt exists in the current directory.
// # testcase: A file named b.json exists. The original file a.txt no longer exists. | `#!/bin/bash`
`echo "Sample content" > a.txt` | `import os`
# Check that b.json exists and a.txt does not
`assert os.path.isfile("b.json"), "Failed: b.json does not exist."`
`assert not os.path.isfile("a.txt"), "Failed: a.txt still exists."`
`print("Passed:File successfully renamed.")` | NA | NA | PRESTO: Correct
GT: Correct |
| Copy file test.txt from directory dir1 to dir2 in the current directory | `#!/bin/bash`
`cp dir1/test.txt dir2/` | # prerequisite: Ensure two directories dir1 and dir2 exist in the current directory, and a file named test.txt exists inside dir1.
# testcase: A file named test.txt exists inside dir2 The content of dir2/test.txt matches that of dir1/test.txt | `#!/bin/bash`
`echo "This is a test file." > test.txt mkdir -p dir2` | `import os`
# Check if file exists in dir2
`assert os.path.isfile("dir2/test.txt"), "Failed: dir2/test.txt does not exist."`
# Check content match
`with open("dir1/test.txt", "r") as f1, open("dir2/test.txt", "r") as f2:`
`    assert f1.read() == f2.read(), "Failed: File contents do not match."`
`print("Passed: File copied correctly.")` | `#!/bin/bash`
`mkdir -p dir1 dir2`
`echo "This is a test file." > dir1/test.txt` | NA | PRESTO: Correct
GT: Correct |
| Print md5 sum of file file.txt in the current directory | `#!/bin/bash`
`md5sum file.txt` | # prerequisite: Ensure there is a file named file.txt in the current directory, containing some known content, so its MD5 checksum can be computed and verified.
# testcase: After running the script, the MD5 hash output of file.txt should match the expected hash of its content. For the content "hello world\n" (i.e., with newline), the correct MD5 hash is:b10a8db164e07.. | `#!/bin/bash`
`echo "hello world" > file.txt` | `import os`
`import hashlib`
`expected_hash = "5eb63bbbe01eeed093cb22bb8f5acdc3"`
`with open("file.txt", "rb") as f:`
`    content = f.read()`
`md5_actual = hashlib.md5(content).hexdigest()`
`assert md5_actual == expected_hash, "Failed: MD5 hash does not match expected value."` | NA | `import hashlib`
# Correct hash for "hello world\n"
`expected_hash = "b10a8db164e07.."`
`with open("file.txt", "rb") as f:`
`    content = f.read()`
`md5_actual = hashlib.md5(content).hexdigest()`
`assert md5_actual == expected_hash, "Test failed: MD5 hash does not match expected value."`
`print("Passed: MD5 hash is correct.")` | PRESTO: Correct
GT: Correct |
| Delete all files in folder 'dir' present in the current directory | `#!/bin/bash`
`rm -f dir/*` | # prerequisite Create a folder named `dir` in the current directory. Add multiple files to the `dir` folder.
# testcase Check if the `dir` folder exists and is empty after execution. | `#!/bin/bash`
`mkdir -p dir`
`touch dir/file1`
`touch dir/file2`
`touch dir/file3` | `import os`
`import sys`
`if not os.listdir('dir'):`
`    print('PASSED: dir is empty')`
`else:`
`    print('FAILED: dir is not empty')`
`    sys.exit(1)` | NA | NA | PRESTO: Correct
GT: Incorrect |
| What is the bash command to create a new systems group called hackers? | `#!/bin/bash`
`groupadd hackers` | # no prerequisite required",
# testcase", Check if a group named "hackers" exists in the system by running the command `getent group hackers` and verifying that the output contains the group name "hackers". | #no prerequisite required | `import subprocess`
`import sys`
# Check if the group 'hackers' exists in the system=
`result = subprocess.run(['getent', 'group', 'hackers'], capture_output=True, text=True)`
`if result.returncode == 0:`
`    print('PASSED: Group "hackers" exists in the system')`
`else:`
`    print('FAILED: Group "hackers" does not exist in the system')`
`    sys.exit(1)` | NA | NA | PRESTO: Correct
GT: Incorrect |

Figure 2: Representative examples demonstrating both successful and failure outcomes of the PRESTO method. The Evaluation column reports the outcome, PRESTO 's predicted label and the ground truth (GT) label.

domain-specific terminology (e.g., Unix account management) prevents the test from enforcing the intended requirement, leading to incorrect verdicts.

These examples show that PRESTO can reliably recover from errors in environment setup and test logic through iterative feedback loops, converging to correct assessments when errors are localizable. At the same time, they reveal two primary failure modes: (i) *coverage gaps* in generated environments, and (ii) *semantic mismatches* in generated tests. Overall, these case studies highlight both the robustness and the current limitations of feedback-driven refinement in automating Bash script evaluation.

# 6 CONCLUSION AND FUTURE WORK

We introduced PRESTO , a modular and execution-aware refinement framework for improving the correctness and robustness of Bash script generation. By leveraging prerequisite-aware test generation and LLM-guided feedback loops, PRESTO moves beyond brittle static evaluation and enables practical, semantic alignment of scripts with their intended functionality. Across two benchmarks, PRESTO consistently outperformed traditional reference-based approaches, demonstrating the value of execution-driven refinement. Unlike methods reliant on gold-standard references—which are often unavailable in deployment settings—PRESTO operates effectively without them, making it well-suited for real-world use. While references remain useful for benchmarking, PRESTO shows that accurate, reference-less evaluation and correction are not only feasible but essential for scalable, production-grade automation. In future work, we aim to integrate PRESTO into CI/CD pipelines to enable ongoing validation of evolving scripts, extend support to other shell dialects and platforms such as Zsh, Fish, and PowerShell, and enhance its environment modeling capabilities by incorporating system-level signals like service health, network state, and container orchestration metadata.

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

## A   REFINEMENT ITERATIONS ANALYSIS

We also analyze the number of refinement iterations required for convergence. Approximately 15% of instances are correctly evaluated without any refinement, reflecting cases where both prerequisite and test generation are accurate in the first attempt. Around 40% of errors are resolved after a single round of refinement, demonstrating the effectiveness of localized corrections. By five iterations, 70–80% of initially incorrect cases are successfully fixed. The remaining instances either stem from flaws in the main script itself or from issues beyond the current capabilities of LLM-based refinement, such as semantic misunderstandings or incomplete coverage that cannot be corrected through additional feedback cycles.

## B   PROMPTS

### B.1   CODE GENERATION

**System**

```
Role:  Bash Scripting Assistant
Objective:
You will receive a natural language task description for functionality that must be
implemented in Bash.  Your job is to return the exact Bash script implementing only
that task - no explanations, no extra text.

Instructions:
# Task:
- Input:  Natural language description of the Bash task to implement.
- Output:  A minimal, correct, POSIX-compliant Bash script.

- Requirements:
  - Only return the Bash code that completes the task.
  - The script must start with '#!/bin/bash' on the first line.
  - Do not include setup, testing, installation, prerequisites, or any unrelated steps.
  - No comments, explanations, logs, or printed messages unless explicitly requested in
the task.
  - You may use 'echo' commands after task execution if needed.
  - Avoid using aliases, advanced Bash-only features, or external dependencies unless
necessary.
  - Do not use 'sudo' or install anything.
  - Ensure the script is deterministic and non-destructive.

- Output Format:
  - Return the final Bash script wrapped in a single code block using '''bash syntax
highlighting.
```

**User**

```
Convert the following task description into only the required Bash code.  Do not
include explanations, comments, or extra output-return just the Bash code that performs
the task.

Task:  {task}

Remember:  Start with #!/bin/bash and provide only the executable script.
```

## B.2 STEPS GENERATION

---

**System**

```
Role:  System Operations Expert
Objective:
Your role is to analyze the provided task description and determine two things:
1.  What minimal prerequisite setup (if any) is required to ensure the task described
can execute successfully.
2.  What testcase checks (if any) can be used to verify that the task described has
been completed successfully by main.sh.

Note:  The output of main.sh is always stored in the variable main_output.txt by
default, so in such cases where comparing values is required you can compare with
main_output.txt content.  So in the #testcase section wherever applicable instead of
running main.sh in testcase use simple main_output.txt content to compare.
While comparing take care of units like for example if the task is to print memory
value then it can be in bytes or KB or MB so extract the numeric part and compare it
with the expected value.  Also in case of dates the date can be in any format even
without `so in short take care of the format.

Instructions:
- The code that performs the task is assumed to be already present in main.sh.
- Your job for prerequisites is to identify only the strictly necessary file or
directory setup (e.g., creating empty files or folders) needed for main.sh to run.
- Do not include setup steps as they are already handled in the script or anything that
involves installing software or packages (assume everything is installed and updated).
- If no prerequisite setup is required, return exactly:
   # no prerequisite required
- Otherwise, list the prerequisites using natural language and wrap them between the
markers # prerequisite and # testcase as shown below.

- For testcases:
   - Analyze the task and list verification steps that confirm if the task completed
successfully.
   - Each check must be written in natural language as a numbered item.
   - Include only post-condition checks (i.e., what to check after main.sh is
executed).
   - Do not repeat any setup steps.
   - If no verification is necessary, return exactly:
    # no tests required
   - Otherwise, wrap the testcase checks inside # testcase markers.

Remember:  Your job is solely to infer environment setup and verification steps
around the existing main.sh implementation which is already there based upon the task
description provided.  Do not rewrite or execute the script itself.
```

---

**User**

```
Task:  {{task}}

Please analyze the above task and return the following:
1.  Prerequisites:  Minimal and strictly necessary file/directory creation steps (if
any) needed to ensure that main.sh can execute the task described.
- Do not include anything already handled by the script.
- Do not include installations or updates (assume everything is already installed).
- If no prerequisites are needed, return exactly:
# no prerequisite required
- Otherwise, wrap the prerequisite steps in a section starting with # prerequisite.

2.  Testcases:  Post-execution checks to verify the task was completed successfully.
- These should be written in natural language as numbered items.
- Only include verification steps (not setup steps).
- Avoid wildcards (*) that might fail expansion.
- If no tests are required, return exactly:  # no tests required
- Otherwise, wrap them in a section starting with # testcase.

Format your response strictly as per the instructions.
```

## B.3 PREREQUISITE GENERATION

---

**System**

```
Role:  Bash Scripting Assistant
Objective:
You will receive natural language prerequisites based on a task already implemented
in `main.sh`.  Your job is to convert these into minimal and strictly necessary
POSIX-compliant Bash commands.

Instructions:
# Prerequisite:
- Input:  Natural language steps.
- Output:  One Bash command per line, using only the following:
   - `mkdir -p <dir>` to create directories
   - `touch <file>` to create empty files

- Restrictions:
   - Do not use variables, wildcards, redirection, piping, `&&`, or `||` and dont
include any explanation.
   - The output must be a single, safe, and non-destructive Bash command line, without
any explanations, comments, or line breaks and start with code with `#!/bin/bash`
   - Do not include commands like `sudo`, `chmod`, or installation commands.
   - Do not create or modify any `.sh` files.
   - Do not include any placeholder content for user data.
   - Avoid wildcards (*) that might fail expansion.
   - Do not invoke `./main.sh` here.

- Format:
   - Wrap the entire prerequisite code block in a single line, prefixed with
`#prerequisite`.
   - If the input is exactly `# no prerequisite required`, output exactly that.
```

---

**User**

```
Convert the following natural language prerequisites into Bash commands as per the
rules:  {{prerequisite_text}}

Example 1:
# prerequisite
1.  Create a file named `log.txt` with 5 lines

Expected Output:
`#prerequisite touch log.txt; echo 'line1' >> log.txt; echo 'line2' >> log.txt; echo
'line3' >> log.txt; echo 'line4' >> log.txt; echo 'line5' >> log.txt`

Example 2:
Input:
`# no prerequisite required`

Expected Output:
`# no prerequisite required`

Example 3:
Input:
# prerequisite
1.  Create two folders named `data` and `results`
2.  Create a text file called `info.txt` inside `data`

Expected Output:
`#prerequisite mkdir -p data; mkdir -p results; touch data/info.txt`
```

## B.4 TESTCASE GENERATION

---

**System**

```
Role:  Python Testing Assistant
Objective:
You will receive natural language test cases and prerequisite information (both text
and Bash code)
based on a task already implemented in `main.sh`.  Your job is to convert only the test
cases into
minimal and strictly necessary multi-line executable Python code for testing.

Note:  The output of `main.sh` is always captured in the variable `main_output.txt`
file.
So in the `#testcase` section, use this `main_output.txt` variable to verify results
instead of re-running `main.sh`.
Also note the values can be numeric or alphanumeric with some text content in it|so if
necessary, extract the numeric or alphanumeric part before comparison.
```

---

864
865
866
867
868
869
870
871
872
873
874
875
876
877
878
879
880
881
882
883
884
885
886
887
888
889
890
891
892
893
894
895
896
897
898
899
900
901
902
903
904
905
906
907
908
909
910
911
912
913
914
915
916
917

**User**

Given the following test cases and prerequisite context, convert the test cases into
multi-line executable Python test code
that validates the output and environment after running the main Bash script.  The
output of the script is stored
in a file `main_output.txt`.

Prerequisite (text + code):
{{prerequisite_text}}
{{prereq_code}}
Test cases:
{{testcase_text}}

Examples:

Example 1:
Input:
Prerequisite (text + code):
1.  Create dir1 and dir2
2.  Create multiple .txt files in dir1
3.  Create multiple non-.txt files in dir1
`#prerequisite mkdir -p dir1; mkdir -p dir2; touch dir1/file1.txt; touch
dir1/file2.txt; touch dir1/image.png; touch dir1/data.csv`
Test cases:
1.  Check if all .txt files created in dir1 are now moved to dir2
Expected Output:
#testcase
import os
import sys
if os.path.isfile('dir2/file1.txt') and os.path.isfile('dir2/file2.txt') and
   not os.path.isfile('dir1/file1.txt') and not os.path.isfile('dir1/file2.txt'):
      print('PASSED: .txt files moved to dir2')
else:
      print('FAILED: .txt files not moved properly')
      sys.exit(1)

Example 2:
...
Example 3:
...
Instructions:
- Prefix the entire code block with a single `#testcase` line at the top
- Do not use assert statements
- The code should be valid Python 3 and executable as a script
- Use `import os`, `import sys`, `filecmp`, `re`, etc., as needed
- Print a clear PASSED or FAILED message and exit with `sys.exit(1)` on failure
- Do not modify or re-generate the `#prerequisite` section | only use it for context

While comparing, take care of units|for example, if the task is to print a memory
value, it can be in bytes, KB, or MB. Extract the numeric part and compare it with the
expected value.

Instructions:
- Input:  Natural language test descriptions.
- Output:  A single line of Python code prefixed with `#testcase`, which performs the
assertion or check.
- Use only:
   - `os.path.isfile(<file>)` to check if a file exists
   - `os.path.isdir(<dir>)` to check if a directory exists
   - `filecmp.cmp(<file1>, <file2>, shallow=False)` to compare files
   - `re.search(<pattern>, open(<file>).read())` to check file contents
- Avoid subprocess-based or shell-like operations

- Format:
   - Wrap the complete check in a single line starting with `#testcase`
   - If a test fails, raise an `AssertionError` or use a Python `assert` statement
   - If no test is required, output exactly:  `# no tests required`
   - Do not generate the `#prerequisite` section | only use it as context if needed

## B.5 PREREQUISITE REFINEMENT

---

### System

```
Role:  Bash Debugging Assistant

Given a merged script with two sections:
- `#prerequisite`:  Defines the environment setup required before executing the
main script.  It typically creates files or directories needed for the script to run
properly.
- `testcase`:  Validates the output or side effects (e.g., file creation, content
change, file movement) of the main script using assertions or checks.

Objective:
Your job is to generate minimal prerequisite Bash code required to fix the given error.
The code should run **before** the main script to ensure it can execute successfully.

Instructions:

1.  **If the script contains a `no prerequisite required` section:**
- If the error is due to the `testcase`, return **only**:
`src code error`
- If the task clearly requires a prerequisite for the script to run (e.g., file or
directory creation), generate the necessary `#prerequisite` section with the exact Bash
commands required, using the section header:
`#prerequisite`
- If no such prerequisite is needed based on the task description, return exactly:
`no prerequisite required`

2.  **If the script contains a `#prerequisite` section:**
- If the error lies within the `#prerequisite` block, fix it and return only the
corrected `#prerequisite` section.
- If the error is in the `testcase`, return exactly:
`src code error`

Response Guidelines:
- Only return Bash code (POSIX-compliant) with appropriate section headers.
- Do **not** include explanations, comments, code blocks, or markdown formatting.
- Ensure the commands are minimal, safe, and directly solve the setup issue without
redundancy.
```

---

**User**

```
The following script failed with this error:
For the task {{test_prompt}}, the following merged script failed to execute properly:
{{merged_script}}
Because of the code snippet inside `#prerequisite`
With error message:
{{agent_last_error}}

Your task is to analyze deeply whether the failure is caused by something in the
`testcase` section or the `#prerequisite` section.

If the error is in the `#prerequisite`, fix **only** that section, and leave the
`testcase` untouched.

**RULES:**
- Do **not** include code blocks (like ``bash), shebangs ('!/bin/bash'), or any extra
lines outside the target section(s).
- Avoid wildcards ('*') that might fail during expansion.
- Do **not** include installation or update commands.  Assume everything is already
installed.
- The fix must be a single, safe, POSIX-compliant Bash command line with **no
comments** or line breaks.
- If the merged script has a `no prerequisite required` section and the error is in the
`testcase`, output **only**:
`src code error`
- If there is **no** `#prerequisite` section and the error clearly indicates missing
prerequisites based on the task, then generate and return the correct prerequisite
section using the header:
`#prerequisite`
- If there **is** a `#prerequisite` section and the error is in that block, fix the
prerequisite and return only:
`#prerequisite` (with corrected commands)
- If the error is due to `testcase` logic even when `#prerequisite` exists, return:
`src code error`

**Section Purposes:**
- `#prerequisite`:  Defines the environment setup needed before running the main code.
It typically creates files, directories, or data required by the script.
- `testcase`:  Validates output or side effects (e.g., file creation, movement, content
changes) after running the main script.

**Code Example 1:**
**Task**:  Move a text file from dir1 to dir2 and verify the move
**Original Script:**
#prerequisite mkdir -p dir1; touch dir1/file.txt
testcase
import os
import sys
if os.path.isfile("dir2/file.txt") and not os.path.isfile("dir1/file.txt"):
print("PASSED: file moved")
else:
print("FAILED: file not moved")
sys.exit(1)
**Error Origin**:  #prerequisite
**Expected Fix Output:**
#prerequisite mkdir -p dir1; mkdir -p dir2; touch dir1/file.txt

Example 2:
...

Example 3:
...

**Error Origin**:  Missing file
**Expected Fix Output:**
src code error
```

## B.6 TESTCASE REFINEMENT

---

**System**

```
Role:  Python-Based Testcase Validator

Objective:
Given a merged script with two sections|
#prerequisite (defines the environment setup required before executing the src code)
testcase (validates the output or side effects of the main script using assertions or
checks)|
|and an error message, identify the failing section and apply the following logic.

Rules:
1.  If the testcase is incorrect, fix only the testcase section.
2.  If the #prerequisite section is incorrect, do NOT fix it.  Instead, return exactly:
src code error
3.  If both sections are correct and the error seems to be due to the main script,
return:
src code error
4.  Output only the modified section(s)|no explanations, comments, or line breaks.
5.  Always include the section header:  'testcase' if you're fixing it.
6.  Never fix or return the #prerequisite section under any condition.
7.  Never include code blocks, shebangs, or extra formatting like '''python.
8.  All testcase fixes must be written in Python using standard libraries (e.g., os,
pathlib, subprocess).
9.  The Python code should validate expected outputs or file system changes performed
by the src code.

Note:
The output of main.sh is always captured in a file named `main_output.txt`.
In the testcase section, use this file to verify results instead of re-running main.sh.
The output may contain numeric or alphanumeric values mixed with text, so extract and
compare the relevant parts if needed.
```

---

1080
1081
1082
1083
1084
1085
1086
1087
1088
1089
1090
1091
1092
1093
1094
1095
1096
1097
1098
1099
1100
1101
1102
1103
1104
1105
1106
1107
1108
1109
1110
1111
1112
1113
1114
1115
1116
1117
1118
1119
1120
1121
1122
1123
1124
1125
1126
1127
1128
1129
1130
1131
1132
1133

**User**

For the task {{test_prompt}} the following merged script failed to execute properly:
{{merged_script}}
Because of the code snippet:  of testcase
With error message:  {{agent_last_error}}

Your task is to fix only the section responsible for the error:  which is `testcase`.

• If the error is clearly due to validation logic or incorrect assertions, fix only the
`testcase` section with fixed multi-line executable Python code.
• If the issue is due to the '#prerequisite', do not fix it.  Instead, respond with:
src code error
• If everything looks correct in both sections and the error is from the main code,
also respond with:
src code error

Important Rules:
1.  Only fix and return the `testcase` section.  Never modify or return the
'#prerequisite' section.
2.  Always preserve and return the `testcase` section header when making changes.
3.  Do not include any explanations, markdown (like ```python), or shebangs.
4.  Testcase fixes must be written in Python using standard libraries like `os`,
`pathlib`, `subprocess`, or `assert`.
5.  Note:  The output of `main.sh` is always captured in a file named
`main_output.txt`.
Use this `main_output.txt` file to verify results instead of re-running `main.sh`.
The output may be numeric or alphanumeric with additional text, so extract and compare
only the relevant part if needed.
6.  The scripts are run sequentially in this order:  `prerequisite.sh`, `main.sh`, then
the `testcase` (Python).
7.  If no clear fix can be determined, respond exactly with:  src code error

Example 1:
For the task 'Check if a file exists and print result' the following merged script
failed to execute properly:
#prerequisite touch data.txt
testcase
import os
if os.path.isfile('data.csv'):
print('PASSED')
else:
print('FAILED')
exit(1)

In the code snippet:  of testcase
With error message:  FAILED: file not found

Expected Output Fix:
testcase
import os
if os.path.isfile('data.txt'):
print('PASSED')
else:
print('FAILED')
exit(1)

Example 2:
...

Example 3:
...

## B.7 MAIN SCRIPT REFINEMENT

**System**

```
Role:  Bash Code Refinement Assistant"
Objective:
Given a natural language task description, an incorrect Bash script, and the
corresponding outputs from the testcase from an automated test, fix the `FAILED` part
of the Bash script so it satisfies the task and passes the test.

Instructions:
Task:
- Input:  A task prompt in natural language, an incorrect Bash script, and Testcase
output checks with FAILED and PASSED messages from a test case.
- Output:  A corrected, minimal, POSIX-compliant Bash script that completes the
described task and resolves the test failure.

- Requirements:
- Only return the corrected Bash script.
- The script must begin with '!/bin/bash` as the first line.
- Do not include explanations, comments, or logs unless explicitly requested.
- Avoid using aliases, non-standard Bash features, or external dependencies unless
necessary.
- Do not include prerequisite setup, installation, or test case logic.
- Do not use `sudo` or install packages.
- Ensure the script is safe, reproducible, and only addresses the described task.

- Output Format:
- Return only the final corrected Bash script inside a single code block using ```bash
syntax highlighting.
```

**User**

```
You are given a task description, a Bash script intended to perform that task, and a
FAILED message from a corresponding test case.
Your job is to FIX the Bash script so that it correctly performs the task and passes
the test.
Task: {test_prompt}
Original Script:  {original_script}
Testcase output with both passed and failed messages:output_message
Original script STDERR:{last_main_err}
Return ONLY the corrected Bash script.
- Start the script with '!/bin/bash`.
- Do not include explanations, comments, or any additional output.
- Only return the minimal working Bash code inside a single code block using ```bash."
```

## C  LLM USAGE

Large Language Models (LLMs) were used solely as an assistive tool for refining the clarity, grammar, and presentation of the writing. They were not involved in research ideation, experimentation, analysis, or in generating any substantive technical contributions.

