# OpenReview forum: "PRESTO: A Framework for Orchestrating System States and Test Cases for Bash Script Verification"
_ICLR.cc/2026/Conference — ICLR 2026 Conference Desk Rejected Submission_

### Official Review · Reviewer_Q5e3 · 2025-10-31

**Soundness:** 2
**Presentation:** 3
**Contribution:** 2
**Rating:** 2
**Confidence:** 5

**Summary:**

This paper targets Bash, which is particularly bad compared to other programming
languages. However, it addresses a very general problem: suppose we have a
prompt for a programming task that we want to use for evaluation (or for RL). We
can easily get a model to output a solution, but it takes a lot of labor to
create an execution environment to run the solution and to write test cases to
evaluate that the solution is correct. The paper attacks this problem by
building an LLM agent-based system that builds both the execution environment
and the test suite.

The system uses feedback to refine the generated code and test suite, which has
an obvious threat to validity: when a failure occurs, it could be because the
environment is wrong (bad precondition), the test suite is wrong (bad
postcondition), or the solution is wrong (bad code). The paper takes steps
to address this so that bugs in the synthesized code do not leak -- i.e., so
that the model doesn't trivialize the test cases to make them pass.

**Strengths:**

- A really important problem
- Likely much larger scope than just Bash (and there are some Python experiments
  in the paper)

**Weaknesses:**

The primary weakness with this paper is that it generates a *single environment*
and *single test script* for each task. (The paper tries to acknowledge this
in Example 4. I assume the ground-truth script is "rm -rf dir".)

We can think of any bash script as a function F, defined as:

    F(fs, env, stdin) = (fs', stdout, stderr)

The limitation of the paper is that it generates a single input tuple `(fs, env,
stdin)` as the environment. The paper shows an example (Example 4), where the
synthesized filesystem fs allows a buggy script to trivially pass the
synthesized test script -- which is a correct test suite.

But, there is a deeper problem, which is that the task may have a logical
disjunction. For example, consider the following task, which is close
to something I regularly do:

> If the hostname is my laptop, load the model from $HOME/models/model_name.
> Otherwise, if the hostname is the cluster, load the module from
> /nfs-shared/my-research-group/models/model_name.

To test that a script implements the behavior described above correctly, it is
not enough to strengthen a test input. Instead, we genuinely must test the
script on two different inputs -- one that simulates the laptop and one that
simulates the cluster.

To address this, the system in the paper would have to be architected quite
differently. The agent would need to plan out several different test
environments simultaneously. In this alternative design, when a failure occurs,
the system would have to consider both strengthening the environment (which it
already does) as well as forking a new test environment (which it does not do).

This is a big change, but I think it is necessary to make the system more
robust.

Minor:

- The Arxiv URL for InverseCoder points to the wrong paper.

**Questions:**

See weaknesses.

---

> ### Author Response · Authors · 2025-11-27
>
> We thank the reviewer for raising this important point about tasks that inherently require
> evaluation under multiple distinct environments. PRESTO’s current design focuses on
> generating a **minimal, single environment and test script per task**, which is sufficient for
> the majority of system-level tasks in existing NL2Bash benchmarks. However, we fully agree
> that there might be tasks that require testing across multiple environments to capture the full
> semantics of the specification.
>
> While PRESTO’s test generator can, in some cases, simulate alternative branches within a
> single test (e.g., by setting different HOSTNAME values inside the test logic), this does not fully
> address tasks that genuinely depend on distinct filesystem or environment states. Supporting
> such cases would require extending PRESTO to **plan, construct, and evaluate multiple
> environments** in parallel, rather than refining a single one.
>
> Importantly, PRESTO’s architecture makes this extension feasible: **environments are
> constructed via a structured Dockerfile**, and incorporating a **multi-scenario planner that
> generates several Dockerfile variants** is a natural next step. This would allow PRESTO to
> fork and explore multiple candidate environments when detecting branching conditions—
> precisely the functionality the reviewer describes.
>
> At the same time, we note that **existing public benchmarks do not contain tasks requiring
> multi-environment simulation**; most failures today arise from **missing prerequisites and
> under-specified tests even in single-scenario settings**. PRESTO’s contribution is to
> significantly improve evaluation accuracy under these practical conditions.
>
> Finally, we acknowledge that certain tasks may remain fundamentally untestable through
> environment simulation alone (e.g., tasks requiring access to a real cluster). For such cases,
> PRESTO includes an **LLM-as-a-judge fallback to provide a complementary, reference-less
> assessment** when executable test generation is not feasible.

---

### Official Review · Reviewer_XrMa · 2025-11-01

**Soundness:** 3
**Presentation:** 2
**Contribution:** 2
**Rating:** 4
**Confidence:** 3

**Summary:**

This paper introduces PRESTO, a framework for the reference-less evaluation of Bash scripts. The authors argue that script verification, especially for LLM-generated code, is fundamentally a problem of state and environment, not just code. PRESTO uses LLMs to generate and iteratively refine two key components from a natural language task: (1) an environmental prerequisite script (P) and (2) a test case script (T). The framework's core novelty is its execution-driven refinement loop, which stabilizes P and T independently of the main script (M) being evaluated. This decoupled design prevents "evaluation leakage" where the environment or test might overfit to a buggy script. Experiments on two benchmarks show PRESTO's evaluation F1-score outperforms execution-less and existing execution-based baselines, and that its feedback signal is uniquely effective for downstream script refinement.

**Strengths:**

The paper tackles a critical and deceptively difficult problem. While much of the community focuses on self-contained, algorithmic code generation (e.g., MBPP, HumanEval), this work correctly identifies that system-level scripts (like Bash) present a distinct and harder challenge due to their profound state-dependency. Addressing the "implicit prerequisite" problem is a novel and valuable contribution.

The core design choice to decompose the problem into Prerequisite (P), Test Case (T), and Main Script (M) is strong. The decision to generate and refine P and T in isolation from M is the paper's strongest methodological contribution. This separation wisely prevents the test harness from being overfit to a specific, potentially buggy, implementation (M), ensuring that P and T remain faithful to the original task specification (S).

 The framework's ability to operate without gold-standard reference scripts is a significant practical advantage. In real-world deployment for SRE or DevOps, such references are non-existent. PRESTO's approach of synthesizing the evaluation criteria (P and T) from the task description is the only scalable path forward.

Value in an Agent-driven World: The framework's value is particularly clear in the context of modern interactive coding agents (e.g., "Cursor-like" tools). While such agents can interact with an environment via trial-and-error, they often lack a systematic oracle for defining correctness. PRESTO provides exactly this: it automates the generation of a minimal, reproducible environment (P) and a formal test (T) that together define the correctness criteria for a given task. This shifts the paradigm from human-guided, trial-and-error debugging to automated, systematic verification. PRESTO's output could, in principle, serve as the "scaffolding" for an interactive agent, providing it with a robust sandbox and a success condition.

**Weaknesses:**

The framework's primary weakness (Examples 4 and 5), is its total reliance on the LLM's "imagination" to generate comprehensive prerequisites and test cases. The evaluation is only as good as the generated tests. A "Coverage Gap" (Example 4) is a critical failure mode: if the LLM fails to generate prerequisites and tests for relevant edge cases (e.g., empty files, non-existent directories, files with spaces, mixed file/directory scenarios), the evaluation will produce false positives. This is a fundamental limitation.

Similarly, the "Semantic Mismatch" (Example 5) is a critical flaw. If the LLM misunderstands a domain-specific term (e.g., "system group" vs. "group"), the entire evaluation framework will be verifying the wrong behavior. The paper does not propose a clear mitigation for these two key failure modes, which are inherent to the LLM-as-generator paradigm.


The extremely poor performance of the adapted AgentCoder baseline (e.g., 31% accuracy with GPT-4o on NL2Bash-EABench) is suspicious. a simpler baseline is to "just run the bash and take the feedback to LLM" - a simple self-debug scaffold, which AgentCoder does not have? I wonder how sota LLM performs. While the authors attribute this to its lack of prerequisite-awareness, the performance is so low that it raises questions about the fairness and fidelity of the adaptation from its original Python domain to Bash.

Given that coverage gaps and semantic ambiguity are the primary failure modes, what extensions to PRESTO are envisioned?

**Questions:**

do you think with the increasing coding ability of LLM, the problem you are tackling will no longer be valid?

---

> ### Author Response · Authors · 2025-11-27
>
> **Coverage gap**
>
> Coverage gaps are an inherent limitation of any LLM-generated test framework.We fully acknowledge that if the LLM fails to imagine certain edge cases (empty files, spaces in filenames, nonexistent directories), the resulting test suite may miss corner-case bugs. This limitation is not unique to PRESTO: it affects all agentic or self-debugging methods that rely on LLMs for generating test stimuli, including ReAct-based agents, AgentCoder, SWE-bench-style scaffolds etc. PRESTO does not claim complete test coverage; instead, its contribution is a **precondition-aware execution-grounded** alternative to purely reference-based or purely heuristic judging methods. PRESTO already improves robustness over LLM-only judging by evaluating concrete execution behaviors rather than relying on LLM semantics alone.
>
> PRESTO **partially mitigates coverage gaps via its iterative refinement loop**. If a missing edge case causes the main script to fail at runtime (e.g., creating a directory that already exists), PRESTO’s refinement loop correctly identifies failures and expands the prerequisite setup accordingly. While this cannot eliminate all coverage gaps, it empirically improves test completeness in many cases. We are careful in the paper not to claim full coverage, and we explicitly identify this limitation in the discussion.
>
> **Semantic Mismatch**
>
> Semantic mismatches arise from natural language ambiguity. PRESTO reduces but cannot fully eliminate them. In PRESTO, test and precondition scripts are generated from the task description, not from the generated Bash code. This reduces the chance of tests being tuned to the model output (which would worsen false positives). But semantic ambiguity in natural language (e.g., “system group”) is a real challenge for all NL2Bash systems. PRESTO’s refinement loop helps catch such mismatch when the script fails under realistic execution, but we agree that deeper semantic grounding remains an open problem.
>
>  **Poor Performance of Agent Coder**
>
> The poor performance is **not an artifact of unfair adaptation**, but stems from a **structural misalignment between AgentCoder’s Python-oriented design and the nature of system-level Bash tasks**. AgentCoder assumes that the execution environment is already valid and that generated tests can be run directly. However, for NL2Bash-EABench and InterCode tasks, **scripts usually depend on implicit environment state** (directories, files, permissions, system users, etc.). Because AgentCoder does not construct or validate prerequisites, the generated tests frequently run in an invalid environment—even when the test logic itself is correct—**leading to execution failures unrelated to the correctness of the generated script**. Under AgentCoder’s decision rules, these failures cause the script to be labeled “incorrect,” explaining the observed drop in accuracy.
>
> This behavior reinforces one of our core findings: For system-level Bash tasks, prerequisite generation and alignment between preconditions and test cases are essential. Without ensuring that the environment matches the task assumptions, even sophisticated agent frameworks (including AgentCoder) systematically mis-evaluate scripts.
>
> We also experimented with the simpler “self-debug” baseline suggested by the reviewer (execute → feed stderr to LLM). As shown in Table 3, this baseline ( same as direct grading)performs better than AgentCoder, but still below PRESTO. This further confirms that **execution alone is insufficient**—what is needed is **execution grounded in properly inferred environment preconditions**.
>
> **Increasing coding ability of LLM**
>
> While stronger LLMs reduce some syntactic errors, they do not eliminate the need for systematic evaluation—especially for Bash and system-level tasks where correctness depends on implicit environment state, file-system conditions, permissions, and side effects that cannot be reliably inferred from text alone. Even if future LLMs generate more accurate code, researchers and practitioners will still require an execution-grounded method to measure how well models generalize across languages, domains, and real-world system conditions. PRESTO targets this structural requirement: without identifying preconditions and executing scripts under aligned test cases, evaluating Bash correctness becomes unreliable and often impossible. Thus, the need for principled, environment-aware evaluation persists regardless of future gains in LLM coding ability.

---

### Official Review · Reviewer_D1A9 · 2025-11-01

**Soundness:** 2
**Presentation:** 2
**Contribution:** 3
**Rating:** 2
**Confidence:** 3

**Summary:**

The paper proposes PRESTO, a prerequisite-aware framework to evaluate and refine Bash scripts generated from natural-language tasks. PRESTO includes (i) environment prerequisite setup, (ii) test-case generation, and (iii) iterative error-attribution and refinement. A key design choice is that prerequisite and test planning never see the main script, reducing leakage and overfitting to a particular implementation.

Experiments on NL2Bash-EABench and InterCode-Corrections compare PRESTO against execution-less evaluators (ICE-Score, Direct Grading) and an execution-based evaluator, AgentCoder. PRESTO achieves high accuracy and macro-F1 across three LLMs and two benchmarks.

**Strengths:**

- **Methodology novelty:**  generation of both prerequisite and tests from the task without seeing the main script.
- **PRESTO Performance**: better macro-F1 than baselines across datasets and models.
- **Case Studies:** highlight cases where PRESTO’s refinement corrects missing prerequisites or brittle tests, as well as failure cases.

**Weaknesses:**

- **Missing Results:**
  - In Table 1, there is no consistent number of decimal points, which makes comparison harder.
  - Report the full table for prerequisite generation performance across models [RQ2].
- **Scalability.** What is the cost / overhead of iterative refinement step? Please add runtime and token accounting.
- **Dependence on LLM-as-judge.** For InterCode-Corrections, using LLM-as-a-Judge for evaluation may mean some “ground truths” are model-derived, not purely executional. This may affect claims about absolute correctness
- **Presentation**
  - Citation format is not consistent, especially in introduction and related works.
  - Typo: testcase $\to$ test case
  - The two paragraphs that start with "Important design note:" in Section 3.2 disrupt the flow.
  - Figure 2 is not readbable due to its font size.

**Questions:**

- How do results change if you use different LLMs for the generator and evaluator?
- Do you use the same LLM for generation and evaluation? It may cause self-confirmation effects.

---

> ### Author Response · Authors · 2025-11-27
>
> **Dependence on LLM-as-judge**
>
> Our use of LLM-as-a-Judge follows the **official InterCode-Corrections evaluation protocol**, which relies on hybrid metrics including TF–IDF, embeddings, and LLM-based semantic checks. In InterCode-Corrections, scripts are **first executed** in Dockerized environments. The **expected output** (from the reference script) is then **compared** against the **generated script’s output**. LLM-as-a-Judge is applied because direct execution-based comparison might be ambiguous as they maybe presented in different ways . For example if the task is to find the total memory of the system, one script may give the memory in mb where as another one may present in in gb. Hence for better comparison, llm is used to determine the **similarity of the outputs after executing both the scripts**.
> We follow this official dataset protocol, not a choice unique to our work.
>
> **Same/Different LLMs for generator and evaluator**
>
> We agree that using the same LLM for code generation and LLM-as-a-Judge evaluation can introduce self-confirmation effects, where the judge may be more lenient toward outputs produced by its own model family. However, our PRESTO-based evaluation substantially mitigates this issue. Unlike LLM-as-Judge methods, PRESTO **never observes the generated Bash code** when constructing preconditions or test cases. All prerequisites and tests are derived solely from the task description, not from the model-generated script. As a result, PRESTO’s evaluation is grounded entirely in **execution-time behavior** (exit codes, stdout/stderr, and file-system effects), rather than subjective model judgments. This execution-grounded approach prevents the kinds of bias commonly observed when a model judges its own outputs. To further validate this, we conducted additional experiments where the **generator and evaluator were different models** (e.g., Llama-4 for generation and GPT-4o for evaluation). We found **no significant performance differences** for PRESTO or for other execution-grounded baselines such as AgentCoder. The only method affected was Direct Grading, where we observed a mild self-confirmation effect: the LLM tended to label outputs from its own family as correct more often. Because these differences did not alter any core trends and the execution-based methods showed no such bias, we did not include these additional cross-model results in the main paper. Overall, PRESTO’s evaluation remains robust regardless of whether the same or different LLMs are used.
>
> **Computational Cost**
>
> In our analysis of PRESTO’s refinement dynamics, we find that the iterative loop is invoked **far less frequently** than the worst case suggests: approximately **50% of tasks require no refinement**, and **65% converge after a single refinement step**. Only **~25%** of tasks reach the maximum of 5 iterations. As a result, the average cost is significantly lower than the theoretical upper bound.
> On average, PRESTO incurs **1.3–1.5 refinement calls per task**. The additional cost arises precisely on tasks where one-shot evaluation is least reliable (e.g., missing prerequisites, under-specified tests), and refinement meaningfully improves correctness. Importantly, the system is tunable: reducing the maximum iteration cap (e.g., from 5 to 2–3) lowers cost substantially with only a small decrease in accuracy.

---

### Official Review · Reviewer_7jC2 · 2025-11-01

**Soundness:** 2
**Presentation:** 3
**Contribution:** 2
**Rating:** 4
**Confidence:** 4

**Summary:**

This paper introduces PRESTO, a modular framework designed to evaluate and refine Bash scripts generated by llms. The core problem it addresses is the unreliability of existing evaluation methods for system-level scripts, which depend heavily on implicit environmental prerequisites. PRESTO's novelty lies in its prerequisite-aware approach: it automatically infers required preconditions, synthesizes minimal environments, generates targeted test cases, and executes an iterative refinement loop—all without relying on gold-standard reference scripts. The framework is evaluated on two NL2Bash benchmarks, showing improvements in evaluation accuracy and enabling more effective script refinement compared to baseline methods.

**Strengths:**

1. The paper identifies the challenge of reliably evaluating and refining system-level scripts where execution context is paramount.
2. The key design choice of separating the generation of prerequisites and test cases from the main script is crucial. This ensures that prerequisites and test cases validate the task specification rather than overfitting to a potentially buggy implementation.
3. The paper provides a thorough experimental evaluation on two complementary benchmarks.
4. The analysis of failure modes in RQ5 is particularly valuable. It provides transparency about the framework's limitations and offers clear directions for future work.

**Weaknesses:**

1. The empirical validation, while solid, is conducted on two relatively small benchmarks. Broader assessment, for instance, on a larger set of language models or more diverse languages, would strengthen the claims of generalizability.
2. The iterative refinement process is inherently more computationally expensive than one-shot methods. A discussion of the computational trade-offs is needed for a complete picture of the framework's practicality.
3. The observation that the "Direct Grading" baseline performs competitively on the NL2Bash-EABench benchmark is notable but under-analyzed. A deeper investigation into why this execution-free method works deceptively well on certain tasks but fails on harder ones would provide valuable insights into the problem's nature and the necessity of PRESTO's approach.

**Questions:**

see weakness

---

> ### Author Response · Authors · 2025-11-27
>
> **On Benchmark Size and Generalizability**
>
> We agree that broader validation across more datasets and languages would further strengthen generalizability. We chose NL2Bash-EABench and InterCode-Corrections because these are the only publicly available, high-quality benchmarks that provide (i) deterministic execution harnesses and (ii) tasks requiring nontrivial environmental prerequisites. Combined they have **~300 diverse examples** covering different aspects of system related tasks with difficulty ranging from easy problems such as moving files to much harder tasks such as creating system groups (which require the llms to understand system related terminologies).
>
> **Why Direct Grading Appears Competitive on NL2Bash-EABench**
>
> NL2Bash-EABench tasks tend to be **shorter, more deterministic, and easier tasks containing fewer environmental dependencies**.
>
> The code generation performance is also very strong for this dataset. Most of the generated code is correct which indicates that the model can perform well on these easier tasks.
>
> As a result, execution-less graders  perform well since the models are capable of performing these tasks.
>
> However, this behavior does not generalize to more complex tasks (as seen in InterCode-Corrections), where Direct Grading degrades sharply.
>
> PRESTO’s advantage is most visible on tasks requiring (i) prerequisite setup, (ii) multi-step reasoning, and (iii) semantic verification—areas where execution-free baselines fundamentally fail.
>
> **Computational Cost**
>
> We agree that iterative refinement introduces additional computation compared to one-shot evaluators. To quantify this, we conducted a detailed analysis of refinement depth across all tasks. We find that **~50% of tasks require no refinement** at all, and **~65% converge after a single refinement iteration**. Only **~25% reach the maximum of 5 iterations**. This indicates that PRESTO’s refinement loop is invoked sparingly and is heavily front-loaded, limiting average-case cost.
> While PRESTO is indeed more expensive than LLM-as-a-judge baselines, this cost corresponds to scenarios where one-shot evaluation is least reliable—precisely the cases where prerequisite errors or weak tests would otherwise lead to incorrect judgments. Furthermore, the computational overhead can be tuned: lowering the maximum iteration cap (e.g., from 5 to 2–3) reduces cost substantially while only modestly affecting accuracy.

---

### Note · Program_Chairs · 2026-01-17
**Submission Desk Rejected by Program Chairs**

The following references in this submission do not refer to real documents and/or have major errors in bibliographic information:

 Xiaopeng Huang, Can Xu, Kazuma Hashimoto, and Caiming Xiong. Language models can selfcorrect: Prompting for debugging with chain-of-thought. arXiv preprint arXiv:2302.12813, 2023b. URL https://arxiv.org/abs/2302.12813.